# Adversarial Symmetric Variational Autoencoder

**Yunchen Pu, Weiyao Wang, Ricardo Henao, Liqun Chen, Zhe Gan, Chunyuan Li
and Lawrence Carin**
Department of Electrical and Computer Engineering, Duke University
`{yp42, ww109, r.henao, lc267, zg27,cl319, lcarin}@duke.edu`

## Abstract

A new form of variational autoencoder (VAE) is developed, in which the joint distribution of data and codes is considered in two (symmetric) forms: ($i$) from observed data fed through the encoder to yield codes, and ($ii$) from latent codes drawn from a simple prior and propagated through the decoder to manifest data. Lower bounds are learned for marginal log-likelihood fits observed data *and* latent codes. When learning with the variational bound, one seeks to minimize the *symmetric* Kullback-Leibler divergence of joint density functions from ($i$) and ($ii$), while simultaneously seeking to maximize the two marginal log-likelihoods. To facilitate learning, a new form of adversarial training is developed. An extensive set of experiments is performed, in which we demonstrate state-of-the-art data reconstruction and generation on several image benchmark datasets.

## 1 Introduction

Recently there has been increasing interest in developing *generative* models of data, offering the promise of learning based on the often vast quantity of unlabeled data. With such learning, one typically seeks to build rich, hierarchical probabilistic models that are able to fit to the distribution of complex real data, and are also capable of realistic data synthesis.

Generative models are often characterized by latent variables (codes), and the variability in the codes encompasses the variation in the data [1, 2]. The generative adversarial network (GAN) [3] employs a generative model in which the code is drawn from a simple distribution (*e.g.*, isotropic Gaussian), and then the code is fed through a sophisticated deep neural network (decoder) to manifest the data. In the context of data synthesis, GANs have shown tremendous capabilities in generating realistic, sharp images from models that learn to mimic the structure of real data [3, 4, 5, 6, 7, 8]. The quality of GAN-generated images has been evaluated by somewhat *ad hoc* metrics like inception score [9].

However, the original GAN formulation does not allow inference of the underlying code, given observed data. This makes it difficult to *quantify* the quality of the generative model, as it is not possible to compute the quality of model fit to data. To provide a principled quantitative analysis of model fit, not only should the generative model synthesize realistic-looking data, one also desires the ability to infer the latent code given data (using an encoder). Recent GAN extensions [10, 11] have sought to address this limitation by learning an inverse mapping (encoder) to project data into the latent space, achieving encouraging results on semi-supervised learning. However, these methods still fail to obtain faithful reproductions of the input data, partly due to model underfitting when learning from a fully adversarial objective [10, 11].

Variational autoencoders (VAEs) are designed to learn both an encoder and decoder, leading to excellent data reconstruction and the ability to quantify a bound on the log-likelihood fit of the model to data [12, 13, 14, 15, 16, 17, 18, 19]. In addition, the inferred latent codes can be utilized in downstream applications, including classification [20] and image captioning [21]. However, new images *synthesized* by VAEs tend to be unspecific and/or blurry, with relatively low resolution. These

limitations of VAEs are becoming increasingly understood. Specifically, the traditional VAE seeks to maximize a lower bound on the log-likelihood of the generative model, and therefore VAEs inherit the limitations of maximum-likelihood (ML) learning [22]. Specifically, in ML-based learning one optimizes the (one-way) Kullback-Leibler (KL) divergence between the distribution of the underlying data and the distribution of the model; such learning does not penalize a model that is capable of generating data that are different from that used for training.

Based on the above observations, it is desirable to build a generative-model learning framework with which one can compute and assess the log-likelihood fit to real (observed) data, while also being capable of generating synthetic samples of high realism. Since GANs and VAEs have complementary strengths, their integration appears desirable, with this a principal contribution of this paper. While integration seems natural, we make important changes to both the VAE and GAN setups, to leverage the best of both. Specifically, we develop a new form of the variational lower bound, manifested jointly for the expected log-likelihood of the observed data *and* for the latent codes. Optimizing this variational bound involves maximizing the expected log-likelihood of the data and codes, while simultaneously minimizing a *symmetric* KL divergence involving the *joint* distribution of data and codes. To compute parts of this variational lower bound, a new form of adversarial learning is invoked. The proposed framework is termed Adversarial Symmetric VAE (AS-VAE), since within the model ($i$) the data and codes are treated in a symmetric manner, ($ii$) a symmetric form of KL divergence is minimized when learning, and ($iii$) adversarial training is utilized. To illustrate the utility of AS-VAE, we perform an extensive set of experiments, demonstrating state-of-the-art data reconstruction and generation on several benchmarks datasets.

## 2 Background and Foundations

Consider an observed data sample $x$, modeled as being drawn from $p_{\theta}(x|z)$, with model parameters $\theta$ and latent code $z$. The prior distribution on the code is denoted $p(z)$, typically a distribution that is easy to draw from, such as isotropic Gaussian. The posterior distribution on the code given data $x$ is $p_{\theta}(z|x)$, and since this is typically intractable, it is approximated as $q_{\phi}(z|x)$, parameterized by learned parameters $\phi$. Conditional distributions $q_{\phi}(z|x)$ and $p_{\theta}(x|z)$ are typically designed such that they are easily sampled and, for flexibility, modeled in terms of neural networks [12]. Since $z$ is a latent code for $x$, $q_{\phi}(z|x)$ is also termed a stochastic *encoder*, with $p_{\theta}(x|z)$ a corresponding stochastic *decoder*. The observed data are assumed drawn from $q(x)$, for which we do not have a explicit form, but from which we have samples, *i.e.*, the ensemble $\{x_i\}_{i=1,N}$ used for learning.

Our goal is to learn the model $p_{\theta}(x) = \int p_{\theta}(x|z)p(z)dz$ such that it synthesizes samples that are well matched to those drawn from $q(x)$. We simultaneously seek to learn a corresponding encoder $q_{\phi}(z|x)$ that is both accurate and efficient to implement. Samples $x$ are synthesized via $x \sim p_{\theta}(x|z)$ with $z \sim p(z)$; $z \sim q_{\phi}(z|x)$ provides an efficient coding of observed $x$, that may be used for other purposes (*e.g.*, classification or caption generation when $x$ is an image [21]).

### 2.1 Traditional Variational Autoencoders and Their Limitations

Maximum likelihood (ML) learning of $\theta$ based on direct evaluation of $p_{\theta}(x)$ is typically intractable. The VAE [12, 13] seeks to bound $p_{\theta}(x)$ by maximizing variational expression $\mathcal{L}_{\text{VAE}}(\theta, \phi)$, with respect to parameters $\{\theta, \phi\}$, where

$$\mathcal{L}_{\text{VAE}}(\theta, \phi) = \mathbb{E}_{q_{\phi}(x,z)} \log \left[ \frac{p_{\theta}(x,z)}{q_{\phi}(z|x)} \right] = \mathbb{E}_{q(x)}[\log p_{\theta}(x) - \text{KL}(q_{\phi}(z|x) \| p_{\theta}(z|x))] \quad (1)$$

$$= -\text{KL}(q_{\phi}(x,z) \| p_{\theta}(x,z)) + \text{const}, \quad (2)$$

with expectations $\mathbb{E}_{q_{\phi}(x,z)}$ and $\mathbb{E}_{q(x)}$ performed approximately via sampling. Specifically, to evaluate $\mathbb{E}_{q_{\phi}(x,z)}$ we draw a finite set of samples $z_i \sim q_{\phi}(z_i|x_i)$, with $x_i \sim q(x)$ denoting the observed data, and for $\mathbb{E}_{q(x)}$, we directly use observed data $x_i \sim q(x)$. When learning $\{\theta, \phi\}$, the expectation using samples from $z_i \sim q_{\phi}(z_i|x_i)$ is implemented via the "reparametrization trick" [12].

Maximizing $\mathcal{L}_{\text{VAE}}(\theta, \phi)$ wrt $\{\theta, \phi\}$ provides a lower bound on $\frac{1}{N} \sum_{i=1}^{N} \log p_{\theta}(x_i)$, hence the VAE setup is an approximation to ML learning of $\theta$. Learning $\theta$ based on $\frac{1}{N} \sum_{i=1}^{N} \log p_{\theta}(x_i)$ is equivalent to learning $\theta$ based on minimizing $\text{KL}(q(x) \| p_{\theta}(x))$, again implemented in terms of the $N$ observed samples of $q(x)$. As discussed in [22], such learning does not penalize $\theta$ severely for yielding $x$

of relatively high probability in $p_{\boldsymbol{\theta}}(\boldsymbol{x})$ while being simultaneously of low probability in $q(\boldsymbol{x})$. This means that $\boldsymbol{\theta}$ seeks to match $p_{\boldsymbol{\theta}}(\boldsymbol{x})$ to the properties of the observed data samples, but $p_{\boldsymbol{\theta}}(\boldsymbol{x})$ may also have high probability of generating samples that do not look like data drawn from $q(\boldsymbol{x})$. This is a fundamental limitation of ML-based learning [22], inherited by the traditional VAE in (1).

One reason for the failing of ML-based learning of $\boldsymbol{\theta}$ is that the cumulative posterior on latent codes $\int p_{\boldsymbol{\theta}}(\boldsymbol{z}|\boldsymbol{x})q(\boldsymbol{x})d\boldsymbol{x} \approx \int q_{\boldsymbol{\phi}}(\boldsymbol{z}|\boldsymbol{x})q(\boldsymbol{x})d\boldsymbol{x} = q_{\boldsymbol{\phi}}(\boldsymbol{z})$ is typically different from $p(\boldsymbol{z})$, which implies that $\boldsymbol{x} \sim p_{\boldsymbol{\theta}}(\boldsymbol{x}|\boldsymbol{z})$, with $\boldsymbol{z} \sim p(\boldsymbol{z})$ may yield samples $\boldsymbol{x}$ that are different from those generated from $q(\boldsymbol{x})$. Hence, when learning $\{\boldsymbol{\theta}, \boldsymbol{\phi}\}$ one may seek to match $p_{\boldsymbol{\theta}}(\boldsymbol{x})$ to samples of $q(\boldsymbol{x})$, as done in (1), while *simultaneously* matching $q_{\boldsymbol{\phi}}(\boldsymbol{z})$ to samples of $p(\boldsymbol{z})$. The expression in (1) provides a variational bound for matching $p_{\boldsymbol{\theta}}(\boldsymbol{x})$ to samples of $q(\boldsymbol{x})$, thus one may naively think to simultaneously set a similar variational expression for $q_{\boldsymbol{\phi}}(\boldsymbol{z})$, with these two variational expressions optimized jointly. However, to compute this additional variational expression we require an analytic expression for $q_{\boldsymbol{\phi}}(\boldsymbol{x}, \boldsymbol{z}) = q_{\boldsymbol{\phi}}(\boldsymbol{z}|\boldsymbol{x})q(\boldsymbol{x})$, which also means we need an analytic expression for $q(\boldsymbol{x})$, which we do not have.

Examining (2), we also note that $\mathcal{L}_{\text{VAE}}(\boldsymbol{\theta}, \boldsymbol{\phi})$ approximates $-\text{KL}(q_{\boldsymbol{\phi}}(\boldsymbol{x}, \boldsymbol{z})\|p_{\boldsymbol{\theta}}(\boldsymbol{x}, \boldsymbol{z}))$, which has limitations aligned with those discussed above for ML-based learning of $\boldsymbol{\theta}$. Analogous to the above discussion, we would also like to consider $-\text{KL}(p_{\boldsymbol{\theta}}(\boldsymbol{x}, \boldsymbol{z})\|q_{\boldsymbol{\phi}}(\boldsymbol{x}, \boldsymbol{z}))$. So motivated, in Section 3 we develop a new form of variational lower bound, applicable to maximizing $\frac{1}{N}\sum_{i=1}^{N}\log p_{\boldsymbol{\theta}}(\boldsymbol{x}_i)$ *and* $\frac{1}{M}\sum_{j=1}^{M}\log q_{\boldsymbol{\phi}}(\boldsymbol{z}_j)$, where $\boldsymbol{z}_j \sim p(\boldsymbol{z})$ is the $j$-th of $M$ samples from $p(\boldsymbol{z})$. We demonstrate that this new framework leverages both $\text{KL}(p_{\boldsymbol{\theta}}(\boldsymbol{x}, \boldsymbol{z})\|q_{\boldsymbol{\phi}}(\boldsymbol{x}, \boldsymbol{z}))$ and $\text{KL}(q_{\boldsymbol{\phi}}(\boldsymbol{x}, \boldsymbol{z})\|p_{\boldsymbol{\theta}}(\boldsymbol{x}, \boldsymbol{z}))$, by extending ideas from adversarial networks.

## 2.2 Adversarial Learning

The original idea of GAN [3] was to build an effective generative model $p_{\boldsymbol{\theta}}(\boldsymbol{x}|\boldsymbol{z})$, with $\boldsymbol{z} \sim p(\boldsymbol{z})$, as discussed above. There was no desire to simultaneously design an inference network $q_{\boldsymbol{\phi}}(\boldsymbol{z}|\boldsymbol{x})$. More recently, authors [10, 11, 23] have devised adversarial networks that seek both $p_{\boldsymbol{\theta}}(\boldsymbol{x}|\boldsymbol{z})$ and $q_{\boldsymbol{\phi}}(\boldsymbol{z}|\boldsymbol{x})$. As an important example, Adversarial Learned Inference (ALI) [10] considers the following objective function:

$$\min_{\boldsymbol{\theta}, \boldsymbol{\phi}} \max_{\boldsymbol{\psi}} \ \mathcal{L}_{\text{ALI}}(\boldsymbol{\theta}, \boldsymbol{\phi}, \boldsymbol{\psi}) = \mathbb{E}_{q_{\boldsymbol{\phi}}(\boldsymbol{x}, \boldsymbol{z})}[\log \sigma(f_{\boldsymbol{\psi}}(\boldsymbol{x}, \boldsymbol{z}))] + \mathbb{E}_{p_{\boldsymbol{\theta}}(\boldsymbol{x}, \boldsymbol{z})}[\log(1 - \sigma(f_{\boldsymbol{\psi}}(\boldsymbol{x}, \boldsymbol{z})))], \quad (3)$$

where the expectations are approximated with samples, as in (1). The function $f_{\boldsymbol{\psi}}(\boldsymbol{x}, \boldsymbol{z})$, termed a *discriminator*, is typically implemented using a neural network with parameters $\boldsymbol{\psi}$ [10, 11]. Note that in (3) we need only sample from $p_{\boldsymbol{\theta}}(\boldsymbol{x}, \boldsymbol{z}) = p_{\boldsymbol{\theta}}(\boldsymbol{x}|\boldsymbol{z})p(\boldsymbol{z})$ and $q_{\boldsymbol{\phi}}(\boldsymbol{x}, \boldsymbol{z}) = q_{\boldsymbol{\phi}}(\boldsymbol{z}|\boldsymbol{x})q(\boldsymbol{x})$, avoiding the need for an explicit form for $q(\boldsymbol{x})$.

The framework in (3) can, in theory, match $p_{\boldsymbol{\theta}}(\boldsymbol{x}, \boldsymbol{z})$ and $q_{\boldsymbol{\phi}}(\boldsymbol{x}, \boldsymbol{z})$, by finding a Nash equilibrium of their respective non-convex objectives [3, 9]. However, training of such adversarial networks is typically based on stochastic gradient descent, which is designed to find a local mode of a cost function, rather than locating an equilibrium [9]. This objective mismatch may lead to the well-known instability issues associated with GAN training [9, 22].

To alleviate this problem, some researchers add a regularization term, such as reconstruction loss [24, 25, 26] or mutual information [4], to the GAN objective, to restrict the space of suitable mapping functions, thus avoiding some of the failure modes of GANs, *i.e.*, mode collapsing. Below we will formally match the joint distributions as in (3), and reconstruction-based regularization will be manifested by generalizing the VAE setup via adversarial learning. Toward this goal we consider the following lemma, which is analogous to Proposition 1 in [3, 23].

**Lemma 1** *Consider Random Variables (RVs) $\boldsymbol{x}$ and $\boldsymbol{z}$ with joint distributions, $p(\boldsymbol{x}, \boldsymbol{z})$ and $q(\boldsymbol{x}, \boldsymbol{z})$. The optimal discriminator $D^*(\boldsymbol{x}, \boldsymbol{z}) = \sigma(f^*(\boldsymbol{x}, \boldsymbol{z}))$ for the following objective*

$$\max_f \ \mathbb{E}_{p(\boldsymbol{x}, \boldsymbol{z})} \log[\sigma(f(\boldsymbol{x}, \boldsymbol{z}))] + \mathbb{E}_{q(\boldsymbol{x}, \boldsymbol{z})}[\log(1 - \sigma(f(\boldsymbol{x}, \boldsymbol{z})))], \quad (4)$$

*is $f^*(\boldsymbol{x}, \boldsymbol{z}) = \log p(\boldsymbol{x}, \boldsymbol{z}) - \log q(\boldsymbol{x}, \boldsymbol{z})$.*

Under Lemma 1, we are able to estimate the $\log q_{\boldsymbol{\phi}}(\boldsymbol{x}, \boldsymbol{z}) - \log p_{\boldsymbol{\theta}}(\boldsymbol{x})p(\boldsymbol{z})$ and $\log p_{\boldsymbol{\theta}}(\boldsymbol{x}, \boldsymbol{z}) - \log q(\boldsymbol{x})q_{\boldsymbol{\phi}}(\boldsymbol{z})$ using the following corollary.

**Corollary 1.1** *For RVs $\boldsymbol{x}$ and $\boldsymbol{z}$ with encoder joint distribution $q_{\boldsymbol{\phi}}(\boldsymbol{x}, \boldsymbol{z}) = q(\boldsymbol{x})q_{\boldsymbol{\phi}}(\boldsymbol{z}|\boldsymbol{x})$ and decoder joint distribution $p_{\boldsymbol{\theta}}(\boldsymbol{x}, \boldsymbol{z}) = p(\boldsymbol{z})p_{\boldsymbol{\theta}}(\boldsymbol{x}|\boldsymbol{z})$, consider the following objectives:*

$$
\begin{aligned}
\max_{\boldsymbol{\psi}_1} \mathcal{L}_{\mathrm{A1}}(\boldsymbol{\psi}_1) &= \mathbb{E}_{\boldsymbol{x} \sim q(\boldsymbol{x}), \boldsymbol{z} \sim q_{\boldsymbol{\phi}}(\boldsymbol{z}|\boldsymbol{x})} \log[\sigma(f_{\boldsymbol{\psi}_1}(\boldsymbol{x}, \boldsymbol{z}))] \\
&\quad + \mathbb{E}_{\boldsymbol{x} \sim p_{\boldsymbol{\theta}}(\boldsymbol{x}|\boldsymbol{z}'), \boldsymbol{z}' \sim p(\boldsymbol{z}), \boldsymbol{z} \sim p(\boldsymbol{z})} [\log(1 - \sigma(f_{\boldsymbol{\psi}_1}(\boldsymbol{x}, \boldsymbol{z})))],
\end{aligned}
\tag{5}
$$

$$
\begin{aligned}
\max_{\boldsymbol{\psi}_2} \mathcal{L}_{\mathrm{A2}}(\boldsymbol{\psi}_2) &= \mathbb{E}_{\boldsymbol{z} \sim p(\boldsymbol{z}), \boldsymbol{x} \sim p_{\boldsymbol{\theta}}(\boldsymbol{x}|\boldsymbol{z})} \log[\sigma(f_{\boldsymbol{\psi}_2}(\boldsymbol{x}, \boldsymbol{z}))] \\
&\quad + \mathbb{E}_{\boldsymbol{z} \sim q_{\boldsymbol{\phi}}(\boldsymbol{z}|\boldsymbol{x}'), \boldsymbol{x}' \sim q(\boldsymbol{x}), \boldsymbol{x} \sim q(\boldsymbol{x})} [\log(1 - \sigma(f_{\boldsymbol{\psi}_2}(\boldsymbol{x}, \boldsymbol{z})))],
\end{aligned}
\tag{6}
$$

*If the parameters $\boldsymbol{\phi}$ and $\boldsymbol{\theta}$ are fixed, with $f_{\boldsymbol{\psi}_1^*}$ the optimal discriminator for (5) and $f_{\boldsymbol{\psi}_2^*}$ the optimal discriminator for (6), then*

$$
f_{\boldsymbol{\psi}_1^*}(\boldsymbol{x}, \boldsymbol{z}) = \log q_{\boldsymbol{\phi}}(\boldsymbol{x}, \boldsymbol{z}) - \log p_{\boldsymbol{\theta}}(\boldsymbol{x})p(\boldsymbol{z}), \quad f_{\boldsymbol{\psi}_2^*}(\boldsymbol{x}, \boldsymbol{z}) = \log p_{\boldsymbol{\theta}}(\boldsymbol{x}, \boldsymbol{z}) - \log q_{\boldsymbol{\phi}}(\boldsymbol{z})q(\boldsymbol{x}). \tag{7}
$$

The proof is provided in the Appendix A. We also assume in Corollary 1.1 that $f_{\boldsymbol{\psi}_1}(\boldsymbol{x}, \boldsymbol{z})$ and $f_{\boldsymbol{\psi}_2}(\boldsymbol{x}, \boldsymbol{z})$ are sufficiently flexible such that there are parameters $\boldsymbol{\psi}_1^*$ and $\boldsymbol{\psi}_2^*$ capable of achieving the equalities in (7). Toward that end, $f_{\boldsymbol{\psi}_1}$ and $f_{\boldsymbol{\psi}_2}$ are implemented as $\boldsymbol{\psi}_1$- and $\boldsymbol{\psi}_2$-parameterized neural networks (details below), to encourage universal approximation [27].

## 3  Adversarial Symmetric Variational Auto-Encoder (AS-VAE)

Consider variational expressions

$$
\mathcal{L}_{\mathrm{VAEx}}(\boldsymbol{\theta}, \boldsymbol{\phi}) = \mathbb{E}_{q(\boldsymbol{x})} \log p_{\boldsymbol{\theta}}(\boldsymbol{x}) - \mathrm{KL}(q_{\boldsymbol{\phi}}(\boldsymbol{x}, \boldsymbol{z})\|p_{\boldsymbol{\theta}}(\boldsymbol{x}, \boldsymbol{z})) \tag{8}
$$

$$
\mathcal{L}_{\mathrm{VAEz}}(\boldsymbol{\theta}, \boldsymbol{\phi}) = \mathbb{E}_{p(\boldsymbol{z})} \log q_{\boldsymbol{\phi}}(\boldsymbol{z}) - \mathrm{KL}(p_{\boldsymbol{\theta}}(\boldsymbol{x}, \boldsymbol{z})\|q_{\boldsymbol{\phi}}(\boldsymbol{x}, \boldsymbol{z})), \tag{9}
$$

where all expectations are again performed approximately using samples from $q(\boldsymbol{x})$ and $p(\boldsymbol{z})$. Recall that $\mathbb{E}_{q(\boldsymbol{x})} \log p_{\boldsymbol{\theta}}(\boldsymbol{x}) = -\mathrm{KL}(q(\boldsymbol{x})\|p_{\boldsymbol{\theta}}(\boldsymbol{x})) + \mathrm{const}$, and $\mathbb{E}_{p(\boldsymbol{z})} \log p_{\boldsymbol{\theta}}(\boldsymbol{z}) = -\mathrm{KL}(p(\boldsymbol{z})\|q_{\boldsymbol{\phi}}(\boldsymbol{z})) + \mathrm{const}$, thus (8) is maximized when $q(\boldsymbol{x}) = p_{\boldsymbol{\theta}}(\boldsymbol{x})$ and $q_{\boldsymbol{\phi}}(\boldsymbol{x}, \boldsymbol{z}) = p_{\boldsymbol{\theta}}(\boldsymbol{x}, \boldsymbol{z})$. Similarly, (9) is maximized when $p(\boldsymbol{z}) = q_{\boldsymbol{\phi}}(\boldsymbol{z})$ and $q_{\boldsymbol{\phi}}(\boldsymbol{x}, \boldsymbol{z}) = p_{\boldsymbol{\theta}}(\boldsymbol{x}, \boldsymbol{z})$. Hence, (8) and (9) impose desired constraints on both the marginal and joint distributions. Note that the log-likelihood terms in (8) and (9) are analogous to the data-fit regularizers discussed above in the context of ALI, but here implemented in a generalized form of the VAE. Direct evaluation of (8) and (9) is not possible, as it requires an explicit form for $q(\boldsymbol{x})$ to evaluate $q_{\boldsymbol{\phi}}(\boldsymbol{x}, \boldsymbol{z}) = q_{\boldsymbol{\phi}}(\boldsymbol{z}|\boldsymbol{x})q(\boldsymbol{x})$.

One may readily demonstrate that

$$
\begin{aligned}
\mathcal{L}_{\mathrm{VAEx}}(\boldsymbol{\theta}, \boldsymbol{\phi}) &= \mathbb{E}_{q_{\boldsymbol{\phi}}(\boldsymbol{x}, \boldsymbol{z})}[\log p_{\boldsymbol{\theta}}(\boldsymbol{x})p(\boldsymbol{z}) - \log q_{\boldsymbol{\phi}}(\boldsymbol{x}, \boldsymbol{z}) + \log p_{\boldsymbol{\theta}}(\boldsymbol{x}|\boldsymbol{z})] \\
&= \mathbb{E}_{q_{\boldsymbol{\phi}}(\boldsymbol{x}, \boldsymbol{z})}[\log p_{\boldsymbol{\theta}}(\boldsymbol{x}|\boldsymbol{z}) - f_{\boldsymbol{\psi}_1^*}(\boldsymbol{x}, \boldsymbol{z})].
\end{aligned}
$$

A similar expression holds for $\mathcal{L}_{\mathrm{VAEz}}(\boldsymbol{\theta}, \boldsymbol{\phi})$, in terms of $f_{\boldsymbol{\psi}_2^*}(\boldsymbol{x}, \boldsymbol{z})$. This naturally suggests the cumulative variational expression

$$
\begin{aligned}
\mathcal{L}_{\mathrm{VAExz}}(\boldsymbol{\theta}, \boldsymbol{\phi}, \boldsymbol{\psi}_1, \boldsymbol{\psi}_2) &= \mathcal{L}_{\mathrm{VAEx}}(\boldsymbol{\theta}, \boldsymbol{\phi}) + \mathcal{L}_{\mathrm{VAEz}}(\boldsymbol{\theta}, \boldsymbol{\phi}) \\
&= \mathbb{E}_{q_{\boldsymbol{\phi}}(\boldsymbol{x}, \boldsymbol{z})}[\log p_{\boldsymbol{\theta}}(\boldsymbol{x}|\boldsymbol{z}) - f_{\boldsymbol{\psi}_1}(\boldsymbol{x}, \boldsymbol{z})] + \mathbb{E}_{p_{\boldsymbol{\theta}}(\boldsymbol{x}, \boldsymbol{z})}[\log q_{\boldsymbol{\phi}}(\boldsymbol{x}|\boldsymbol{z}) - f_{\boldsymbol{\psi}_2}(\boldsymbol{x}, \boldsymbol{z})],
\end{aligned}
\tag{10}
$$

where $\boldsymbol{\psi}_1$ and $\boldsymbol{\psi}_2$ are updated using the adversarial objectives in (5) and (6), respectively.

Note that to evaluate (10) we must be able to sample from $q_{\boldsymbol{\phi}}(\boldsymbol{x}, \boldsymbol{z}) = q(\boldsymbol{x})q_{\boldsymbol{\phi}}(\boldsymbol{z}|\boldsymbol{x})$ and $p_{\boldsymbol{\theta}}(\boldsymbol{x}, \boldsymbol{z}) = p(\boldsymbol{z})p_{\boldsymbol{\theta}}(\boldsymbol{x}|\boldsymbol{z})$, both of which are readily available, as discussed above. Further, we require explicit expressions for $q_{\boldsymbol{\phi}}(\boldsymbol{z}|\boldsymbol{x})$ and $p_{\boldsymbol{\theta}}(\boldsymbol{x}|\boldsymbol{z})$, which we have. For (5) and (6) we similarly must be able to sample from the distributions involved, and we must be able to evaluate $f_{\boldsymbol{\psi}_1}(\boldsymbol{x}, \boldsymbol{z})$ and $f_{\boldsymbol{\psi}_2}(\boldsymbol{x}, \boldsymbol{z})$, each of which is implemented via a neural network. Note as well that the bound in (1) for $\mathbb{E}_{q(\boldsymbol{x})} \log p_{\boldsymbol{\theta}}(\boldsymbol{x})$ is in terms of the KL distance between *conditional* distributions $q_{\boldsymbol{\phi}}(\boldsymbol{z}|\boldsymbol{x})$ and $p_{\boldsymbol{\theta}}(\boldsymbol{z}|\boldsymbol{x})$, while (8) utilizes the KL distance between *joint* distributions $q_{\boldsymbol{\phi}}(\boldsymbol{x}, \boldsymbol{z})$ and $p_{\boldsymbol{\theta}}(\boldsymbol{x}, \boldsymbol{z})$ (use of joint distributions is related to ALI). By combining (8) and (9), the complete variational bound $\mathcal{L}_{\mathrm{VAExz}}$ employs the *symmetric* KL between these two joint distributions. By contrast, from (2), the original variational lower bound only addresses a *one-way* KL distance between $q_{\boldsymbol{\phi}}(\boldsymbol{x}, \boldsymbol{z})$ and $p_{\boldsymbol{\theta}}(\boldsymbol{x}, \boldsymbol{z})$. While [23] had a similar idea of employing adversarial methods in the context variational learning, it was only done within the context of the original form in (1), the limitations of which were discussed in Section 2.1.

In the original VAE, in which (1) was optimized, the reparametrization trick [12] was invoked wrt $q_\phi(z|x)$, with samples $z_\phi(x, \epsilon)$ and $\epsilon \sim \mathcal{N}(0, \mathbf{I})$, as the expectation was performed wrt this distribution; this reparametrization is convenient for computing gradients wrt $\phi$. In the AS-VAE in (10), expectations are also needed wrt $p_\theta(x|z)$. Hence, to implement gradients wrt $\theta$, we also constitute a reparametrization of $p_\theta(x|z)$. Specifically, we consider samples $x_\theta(z, \xi)$ with $\xi \sim \mathcal{N}(0, \mathbf{I})$. $\mathcal{L}_{\text{VAExz}}(\theta, \phi, \psi_1, \psi_2)$ in (10) is re-expressed as

$$\mathcal{L}_{\text{VAExz}}(\theta, \phi, \psi_1, \psi_2) = \mathbb{E}_{x \sim q(x), \epsilon \sim \mathcal{N}(0,\mathbf{I})} \left[ f_{\psi_1}(x, z_\phi(x, \epsilon)) - \log p_\theta(x|z_\phi(x, \epsilon)) \right]$$
$$+ \mathbb{E}_{z \sim p(z), \xi \sim \mathcal{N}(0,\mathbf{I})} \left[ f_{\psi_2}(x_\theta(z, \xi), z) - \log q_\phi(z|x_\theta(z, \xi)) \right]. \quad (11)$$

The expectations in (11) are approximated via samples drawn from $q(x)$ and $p(z)$, as well as samples of $\epsilon$ and $\xi$. $x_\theta(z, \xi)$ and $z_\phi(x, \epsilon)$ can be implemented with a Gaussian assumption [12] or via density transformation [14, 16], detailed when presenting experiments in Section 5.

The complete objective of the proposed Adversarial Symmetric VAE (AS-VAE) requires the cumulative variational in (11), which we maximize wrt $\psi_1$ and $\psi_1$ as in (5) and (6), using the results in (7). Hence, we write

$$\min_{\theta, \phi} \max_{\psi_1, \psi_2} \quad -\mathcal{L}_{\text{VAExz}}(\theta, \phi, \psi_1, \psi_2). \quad (12)$$

The following proposition characterizes the solutions of (12) in terms of the joint distributions of $x$ and $z$.

**Proposition 1** *The equilibrium for the min-max objective in* (12) *is achieved by specification* $\{\theta^*, \phi^*, \psi_1^*, \psi_2^*\}$ *if and only if* (7) *holds, and* $p_{\theta^*}(x, z) = q_{\phi^*}(x, z)$.

The proof is provided in the Appendix A. This theoretical result implies that $(i)$ $\theta^*$ is an estimator that yields good reconstruction, and $(ii)$ $\phi^*$ matches the aggregated posterior $q_\phi(z)$ to prior distribution $p(z)$.

# 4 Related Work

VAEs [12, 13] represent one of the most successful deep generative models developed recently. Aided by the reparameterization trick, VAEs can be trained with stochastic gradient descent. The original VAEs implement a Gaussian assumption for the encoder. More recently, there has been a desire to remove this Gaussian assumption. Normalizing flow [14] employs a sequence of invertible transformation to make the distribution of the latent codes arbitrarily flexible. This work was followed by inverse auto-regressive flow [16], which uses recurrent neural networks to make the latent codes more expressive. More recently, SteinVAE [28] applies Stein variational gradient descent [29] to infer the distribution of latent codes, discarding the assumption of a parametric form of posterior distribution for the latent code. However, these methods are not able to address the fundamental limitation of ML-based models, as they are all based on the variational formulation in (1).

GANs [3] constitute another recent framework for learning a generative model. Recent extensions of GAN have focused on boosting the performance of image generation by improving the generator [5], discriminator [30] or the training algorithm [9, 22, 31]. More recently, some researchers [10, 11, 33] have employed a bidirectional network structure within the adversarial learning framework, which in theory guarantees the matching of joint distributions over two domains. However, non-identifiability issues are raised in [32]. For example, they have difficulties in providing good reconstruction in latent variable models, or discovering the correct pairing relationship in domain transformation tasks. It was shown that these problems are alleviated in DiscoGAN [24], CycleGAN [26] and ALICE [32] via additional $\ell_1$, $\ell_2$ or adversarial losses. However, these methods lack of explicit probabilistic modeling of observations, thus could not directly evaluate the likelihood of given data samples.

A key component of the proposed framework concerns integrating a new VAE formulation with adversarial learning. There are several recent approaches that have tried to combining VAE and GAN [34, 35], Adversarial Variational Bayes (AVB) [23] is the one most closely related to our work. AVB employs adversarial learning to estimate the posterior of the latent codes, which makes the encoder arbitrarily flexible. However, AVB seeks to optimize the original VAE formulation in (1), and hence it inherits the limitations of ML-based learning of $\theta$. Unlike AVB, the proposed use of adversarial learning is based on a new VAE setup, that seeks to minimize the *symmetric* KL distance between $p_\theta(x, z)$ and $q_\phi(x, z)$, while simultaneously seeking to maximize the marginal expected likelihoods $\mathbb{E}_{q(x)}[\log p_\theta(x)]$ and $\mathbb{E}_{p(z)}[\log p_\phi(z)]$.

# 5 Experiments

We evaluate our model on three datasets: MNIST, CIFAR-10 and ImageNet. To balance performance and computational cost, $p_{\boldsymbol{\theta}}(\boldsymbol{x}|\boldsymbol{z})$ and $q_{\boldsymbol{\phi}}(\boldsymbol{z}|\boldsymbol{x})$ are approximated with a normalizing flow [14] of length 80 for the MNIST dataset, and a Gaussian approximation for CIFAR-10 and ImageNet data. All network architectures are provided in the Appendix B. All parameters were initialized with Xavier [36], and optimized via Adam [37] with learning rate 0.0001. We do not perform any dataset-specific tuning or regularization other than dropout [38]. Early stopping is employed based on average reconstruction loss of $\boldsymbol{x}$ and $\boldsymbol{z}$ on validation sets.

We show three types of results, using part of or all of our model to illustrate each component. $i$) *AS-VAE-r*: This model trained with the first half of the objective in (11) to minimize $\mathcal{L}_{\text{VAEx}}(\boldsymbol{\theta}, \boldsymbol{\phi})$ in (8); it is an ML-based method which focuses on reconstruction. $ii$) *AS-VAE-g*: This model trained with the second half of the objective in (11) to minimize $\mathcal{L}_{\text{VAEz}}(\boldsymbol{\theta}, \boldsymbol{\phi})$ in (9); it can be considered as maximizing the likelihood of $q_{\boldsymbol{\phi}}(\boldsymbol{z})$, and designed for generation. $iii$) *AS-VAE* This is our proposed model, developed in Section 3.

## 5.1 Evaluation

We evaluate our model on both reconstruction and generation. The performance of the former is evaluated using negative log-likelihood (NLL) estimated via the variational lower bound defined in (1). Images are modeled as continuous. To do this, we add $[0, 1]$-uniform noise to natural images (one color channel at the time), then divide by 256 to map 8-bit images (256 levels) to the unit interval. This technique is widely used in applications involving natural images [12, 14, 16, 39, 40], since it can be proved that in terms of log-likelihood, modeling in the discrete space is equivalent to modeling in the continuous space (with added noise) [39, 41]. During testing, the likelihood is computed as $p(x = i|\boldsymbol{z}) = p_{\boldsymbol{\theta}}(x \in [i/256, (i + 1)/256]|\boldsymbol{z})$ where $i = 0, \ldots, 255$. This is done to guarantee a fair comparison with prior work (that assumed quantization). For the MNIST dataset, we treat the $[0, 1]$-mapped continuous input as the probability of a binary pixel value (on or off) [12]. The inception score (IS), defined as $\exp(\mathbb{E}_q(\boldsymbol{x}) \text{KL}(p(y|\boldsymbol{x})\|p(y)))$, is employed to quantitatively evaluate the quality of *generated* natural images, where $p(y)$ is the empirical distribution of labels (we do *not* leverage any label information during training) and $p(y|\boldsymbol{x})$ is the output of the Inception model [42] on each generated image.

To the authors' knowledge, we are the first to report both inception score (IS) and NLL for natural images from a single model. For comparison, we implemented DCGAN [5] and PixelCNN++ [40] as baselines. The implementation of DCGAN is based on a similar network architectures as our model. Note that for NLL a lower value is better, whereas for IS a higher value is better.

## 5.2 MNIST

We first evaluate our model on the MNIST dataset. The log-likelihood results are summarized in Table 1. Our AS-VAE achieves a negative log-likelihood of 82.51 nats, outperforming normalizing flow (85.1 nats) with a similar architecture. The perfomance of AS-VAE-r (81.14 nats) is competitive to the state-of-the-art (79.2 nats). The generated samples are showed in Figure 1. AS-VAE-g and AS-VAE both generate good samples while the results of AS-VAE-r are slightly more blurry, partly due to the fact that AS-VAE-r is an ML-based model.

## 5.3 CIFAR

Next we evaluate our models on the CIFAR-10 dataset. The quantitative results are listed in Table 2. AS-VAE-r and AS-VAE-g achieve encouraging results on reconstruction and generation, respectively, while our AS-VAE model (leveraging the full objective) achieves a good balance between these two tasks, which demonstrates the benefit of optimizing a *symmetric* objective. Compared with

Table 1: NLL on MNIST.

| Method | NF (k=80) [14] | IAF [16] | AVB [23] | PixelRNN [39] | AS-VAE-r | AS-VAE-g | AS-VAE |
|---|---|---|---|---|---|---|---|
| **NLL (nats)** | 85.1 | 80.9 | 79.5 | 79.2 | 81.14 | 146.32 | 82.51 |

state-of-the-art ML-based models [39, 40], we achieve competitive results on reconstruction but provide a much better performance on generation, also outperforming other adversarially-trained models. Note that our negative ELBO (evidence lower bound) is an upper bound of NLL as reported in [39, 40]. We also achieve a smaller root-mean-square-error (RMSE). Generated samples are shown in Figure 2. Additional results are provided in the Appendix C.

ALI [10], which also seeks to match the joint encoder and decoder distribution, is also implemented as a baseline. Since the decoder in ALI is a deterministic network, the NLL of ALI is impractical to compute. Alternatively, we report the RMSE of reconstruction as a reference. Figure 3 qualitatively compares the reconstruction performance of our model, ALI and VAE. As can be seen, the reconstruction of ALI is related to but not faithful reproduction of the input data, which evidences the limitation in reconstruction ability of adversarial learning. This is also consistent in terms of RMSE.

Table 2: Quantitative Results on CIFAR-10; [†] 2.96 is based on our implementation and 2.92 is reported in [40].

| Method | NLL(bits) | RMSE | IS |
|---|---|---|---|
| WGAN [43] | - | - | 3.82 |
| MIX+WassersteinGAN [43] | - | - | 4.05 |
| DCGAN [5] | - | - | 4.89 |
| ALI | - | 14.53 | 4.79 |
| PixelRNN [39] | 3.06 | - | - |
| PixelCNN++ [40] | $2.96 (2.92)^{\dagger}$ | 3.289 | 5.51 |
| AS-VAE-r | 3.09 | 3.17 | 2.91 |
| AS-VAE-g | 93.12 | 13.12 | 6.89 |
| AS-VAE | 3.32 | 3.36 | 6.34 |

## 5.4 ImageNet

ImageNet 2012 is used to evaluate the scalability of our model to large datasets. The images are resized to $64 \times 64$. The quantitative results shown in Table 3. Our model significantly improves the performance on generation compared with DCGAN and PixelCNN++, while achieving competitive results on reconstruction compared with PixelRNN and PixelCNN++.

Note that the PixelCNN++ takes more than two weeks (44 hours per epoch) for training and 52.0 seconds/image for generating samples while our model only requires less than 2 days (4 hours per epoch) for training and 0.01 seconds/image for generating on a single TITAN X GPU. As a reference, the true validation set of ImageNet 2012 achieves $53.24\%$ accuracy. This is because ImageNet has much greater variety of images than CIFAR-10. Figure 4 shows generated samples based on trained with ImageNet, compared with DCGAN and PixelCNN++. Our model is able

Table 3: Quantitative Results on ImageNet.

| Method | NLL | IS |
|---|---|---|
| DCGAN [5] | - | 5.965 |
| PixelRNN [39] | 3.63 | - |
| PixelCNN++ [40] | 3.27 | 7.65 |
| AS-VAE | 3.71 | 11.14 |

to produce sharp images without label information while capturing more local spatial dependencies than PixelCNN++, and without suffering from mode collapse as DCGAN. Additional results are provided in the Appendix C.

## 6 Conclusions

We presented Adversarial Symmetrical Variational Autoencoders, a novel deep generative model for unsupervised learning. The learning objective is to minimizing a symmetric KL divergence between the joint distribution of data and latent codes from encoder and decoder, while simultaneously maximizing the expected marginal likelihood of data and codes. An extensive set of results demonstrated excellent performance on both reconstruction and generation, while scaling to large datasets. A possible direction for future work is to apply AS-VAE to semi-supervised learning tasks.

## Acknowledgements

This research was supported in part by ARO, DARPA, DOE, NGA, ONR and NSF.

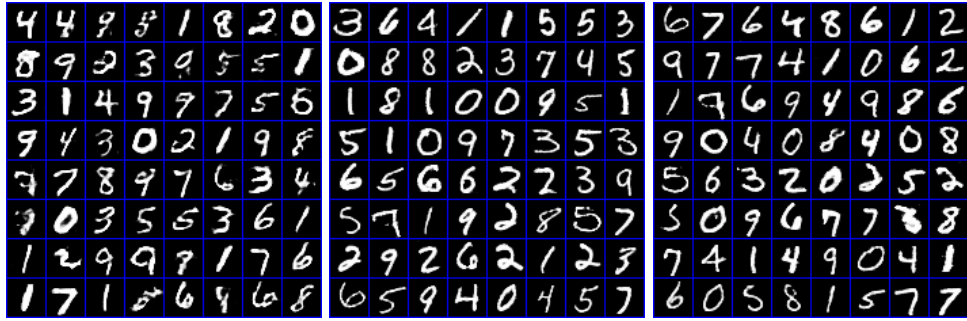

Figure 1: Generated samples trained on MNIST. (Left) AS-VAE-r; (Middle) AS-VAE-g (Right) AS-VAE.

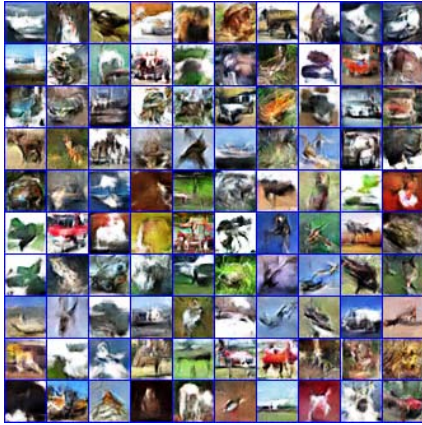

Figure 2: Samples generated by AS-VAE when trained on CIFAR-10.

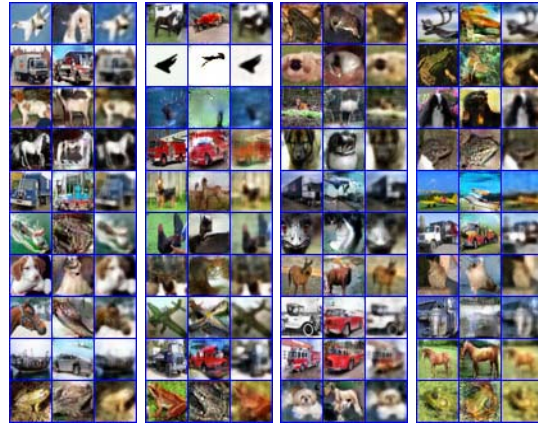

Figure 3: Comparison of reconstruction with ALI [10]. In each block: column one for ground-truth, column two for ALI and column three for AS-VAE.

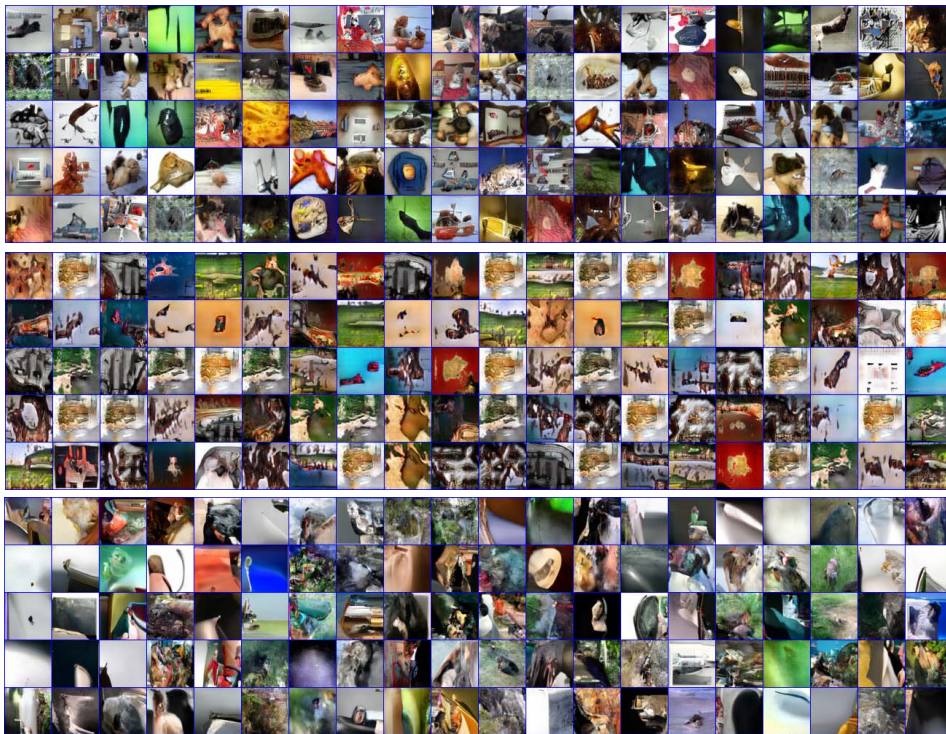

Figure 4: Generated samples trained on ImageNet. (Top) AS-VAE; (Middle) DCGAN [5];(Bottom) Pixel-CNN++ [40].

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
