[Supplementary Material]

# Appendix for "Adversarial Symmetric Variational Autoencoder"

**Yunchen Pu, Weiyao Wang, Ricardo Henao, Liqun Chen, Zhe Gan, Chunyuan Li
and Lawrence Carin**
Department of Electrical and Computer Engineering, Duke University
{yp42, ww109, r.henao, lc267, zg27,cl319, lcarin}@duke.edu

## A  Proof

**Proof of Corollary 1.1**  We start from a simple observation $p_{\boldsymbol{\theta}}(\boldsymbol{x}) = \int_{\boldsymbol{z}} p_{\boldsymbol{\theta}}(\boldsymbol{x}, \boldsymbol{z})d\boldsymbol{z} = \int_{\boldsymbol{z}} p(\boldsymbol{z})p_{\boldsymbol{\theta}}(\boldsymbol{x}|\boldsymbol{z})d\boldsymbol{z}$. The second term in (5) of main paper can be rewritten as

$$\mathbb{E}_{\boldsymbol{x}\sim p_{\boldsymbol{\theta}}(\boldsymbol{x}|\boldsymbol{z}'),\boldsymbol{z}'\sim p(\boldsymbol{z}),\boldsymbol{z}\sim p(\boldsymbol{z})}[\log(1 - \sigma(f_{\boldsymbol{\psi}_1}(\boldsymbol{x}, \boldsymbol{z})))], \tag{1}$$

$$= \int_{\boldsymbol{x}} \int_{\boldsymbol{z}'} \int_{\boldsymbol{z}} p_{\boldsymbol{\theta}}(\boldsymbol{x}|\boldsymbol{z}')p(\boldsymbol{z}')p(\boldsymbol{z}) \log(1 - \sigma(f_{\boldsymbol{\psi}_1}(\boldsymbol{x}, \boldsymbol{z})))d\boldsymbol{x}d\boldsymbol{z}d\boldsymbol{z}' \tag{2}$$

$$= \int_{\boldsymbol{x}} \int_{\boldsymbol{z}} \left\{ \int_{\boldsymbol{z}'} p_{\boldsymbol{\theta}}(\boldsymbol{x}|\boldsymbol{z}')p(\boldsymbol{z}')d\boldsymbol{z}' \right\} p(\boldsymbol{z}) \log(1 - \sigma(f_{\boldsymbol{\psi}_1}(\boldsymbol{x}, \boldsymbol{z})))d\boldsymbol{x}d\boldsymbol{z} \tag{3}$$

$$= \int_{\boldsymbol{x}} \int_{\boldsymbol{z}} p_{\boldsymbol{\theta}}(\boldsymbol{x})p(\boldsymbol{z}) \log(1 - \sigma(f_{\boldsymbol{\psi}_1}(\boldsymbol{x}, \boldsymbol{z})))d\boldsymbol{x}d\boldsymbol{z} \tag{4}$$

Therefore, the objective function $\mathcal{L}_{\mathrm{A1}}(\boldsymbol{\psi}_1)$ in (5) can be expressed as

$$\int_{\boldsymbol{x}} \int_{\boldsymbol{z}} q(\boldsymbol{x})q_{\boldsymbol{\phi}}(\boldsymbol{z}|\boldsymbol{x}) \log[\sigma(f_{\boldsymbol{\psi}_1}(\boldsymbol{x}, \boldsymbol{z}))]d\boldsymbol{x}d\boldsymbol{z} + \int_{\boldsymbol{x}} \int_{\boldsymbol{z}} p_{\boldsymbol{\theta}}(\boldsymbol{x})p(\boldsymbol{z}) \log(1 - \sigma(f_{\boldsymbol{\psi}_1}(\boldsymbol{x}, \boldsymbol{z})))d\boldsymbol{x}d\boldsymbol{z}$$
$$= \int_{\boldsymbol{x}} \int_{\boldsymbol{z}} \left\{ q_{\boldsymbol{\phi}}(\boldsymbol{x}, \boldsymbol{z}) \log[\sigma(f_{\boldsymbol{\psi}_1}(\boldsymbol{x}, \boldsymbol{z}))] + p_{\boldsymbol{\theta}}(\boldsymbol{x})p(\boldsymbol{z}) \log(1 - \sigma(f_{\boldsymbol{\psi}_1}(\boldsymbol{x}, \boldsymbol{z}))) \right\} d\boldsymbol{x}d\boldsymbol{z} \tag{5}$$

This integral of (5) is maximal as a function of $f(\boldsymbol{x}, \boldsymbol{z})$ if and only if the integrand is maximal for every $(\boldsymbol{x}, \boldsymbol{z})$. Note that the problem $max_x a \log x + b \log(1-x)$ achieves maximum at $x = a/(a+b)$ and $\sigma(x) = 1/(1 + e^{-x})$. Hence, we have the optimal function of $f_{\boldsymbol{\psi}_1}$ at

$$\sigma(f_{\boldsymbol{\psi}_1^*}) = \frac{q_{\boldsymbol{\phi}}(\boldsymbol{x}, \boldsymbol{z})}{q_{\boldsymbol{\phi}}(\boldsymbol{x}, \boldsymbol{z}) + p_{\boldsymbol{\theta}}(\boldsymbol{x})p(\boldsymbol{z})} \qquad f_{\boldsymbol{\psi}_1^*} = \log q_{\boldsymbol{\phi}}(\boldsymbol{x}, \boldsymbol{z}) + \log p_{\boldsymbol{\theta}}(\boldsymbol{x})p(\boldsymbol{z}) \tag{6}$$

Similarly, we have $f_{\boldsymbol{\psi}_2^*}(\boldsymbol{x}, \boldsymbol{z}) = \log p_{\boldsymbol{\theta}}(\boldsymbol{x}, \boldsymbol{z}) - \log q_{\boldsymbol{\phi}}(\boldsymbol{z})q(\boldsymbol{x})$

**Proof of Proposition 1**  If $\{\boldsymbol{\theta}^*, \boldsymbol{\phi}^*, \boldsymbol{\psi}_1^*, \boldsymbol{\psi}_2^*\}$ achieves an equilibrium of (12) of main paper. The Corollary 1.1 indicates that $f_{\boldsymbol{\psi}_1^*} = \log q_{\boldsymbol{\phi}}(\boldsymbol{x}, \boldsymbol{z}) + \log p_{\boldsymbol{\theta}}(\boldsymbol{x})p(\boldsymbol{z})$ and $f_{\boldsymbol{\psi}_2^*}(\boldsymbol{x}, \boldsymbol{z}) = \log p_{\boldsymbol{\theta}}(\boldsymbol{x}, \boldsymbol{z}) - \log q_{\boldsymbol{\phi}}(\boldsymbol{z})q(\boldsymbol{x})$.

Note that

$$\mathcal{L}_{\mathrm{VAEx}}(\boldsymbol{\theta}, \boldsymbol{\phi}) = \mathbb{E}_{q(\boldsymbol{x})} \log p_{\boldsymbol{\theta}}(\boldsymbol{x}) - \mathrm{KL}(q_{\boldsymbol{\phi}}(\boldsymbol{x}, \boldsymbol{z})\|p_{\boldsymbol{\theta}}(\boldsymbol{x}, \boldsymbol{z})) \tag{7}$$

$$= \mathbb{E}_{q(\boldsymbol{x})} \log q(\boldsymbol{x}) - \mathrm{KL}(q_{\boldsymbol{\phi}}(\boldsymbol{x}, \boldsymbol{z})\|p_{\boldsymbol{\theta}}(\boldsymbol{x}, \boldsymbol{z})) - \mathrm{KL}(q_{\boldsymbol{\phi}}(\boldsymbol{x})\|p_{\boldsymbol{\theta}}(\boldsymbol{x})) \tag{8}$$

and

$$\mathcal{L}_{\mathrm{VAEz}}(\boldsymbol{\theta}, \boldsymbol{\phi}) = \mathbb{E}_{p(\boldsymbol{z})} \log q_{\boldsymbol{\phi}}(\boldsymbol{z}) - \mathrm{KL}(p_{\boldsymbol{\theta}}(\boldsymbol{x}, \boldsymbol{z})\|q_{\boldsymbol{\phi}}(\boldsymbol{x}, \boldsymbol{z})) \tag{9}$$

$$= \mathbb{E}_{p(\boldsymbol{z})} \log p(\boldsymbol{z}) - \mathrm{KL}(p_{\boldsymbol{\theta}}(\boldsymbol{x}, \boldsymbol{z})\|q_{\boldsymbol{\phi}}(\boldsymbol{x}, \boldsymbol{z})) - \mathrm{KL}(p_{\boldsymbol{\theta}}(\boldsymbol{z})\|q_{\boldsymbol{\phi}}(\boldsymbol{z})) \tag{10}$$

where $\mathbb{E}_{p(\boldsymbol{z})} \log p(\boldsymbol{z})$ and $\mathbb{E}_{q(\boldsymbol{x})} \log q(\boldsymbol{x})$ can be considered as constant. Therefore, maximize $\mathcal{L}_{\text{VAExz}}$ is equivalent to minimize

$$\text{KL}(p_{\boldsymbol{\theta}}(\boldsymbol{x}, \boldsymbol{z}) \| q_{\boldsymbol{\phi}}(\boldsymbol{x}, \boldsymbol{z})) + \text{KL}(q_{\boldsymbol{\phi}}(\boldsymbol{x}, \boldsymbol{z}) \| p_{\boldsymbol{\theta}}(\boldsymbol{x}, \boldsymbol{z})) + \text{KL}(p_{\boldsymbol{\theta}}(\boldsymbol{z}) \| q_{\boldsymbol{\phi}}(\boldsymbol{z})) + \text{KL}(q_{\boldsymbol{\phi}}(\boldsymbol{x}) \| p_{\boldsymbol{\theta}}(\boldsymbol{x}))$$

The minimum of first two terms is achieved if and only if $p_{\boldsymbol{\theta}}(\boldsymbol{x}, \boldsymbol{z}) = q_{\boldsymbol{\phi}}(\boldsymbol{x}, \boldsymbol{z})$ while the minimums of last two terms are achieved at $p_{\boldsymbol{\theta}}(\boldsymbol{x}) = q(\boldsymbol{x})$ and $p(\boldsymbol{z}) = q_{\boldsymbol{\phi}}(\boldsymbol{z})$, respectively. Note that the joint match $p_{\boldsymbol{\theta}}(\boldsymbol{x}, \boldsymbol{z}) = q_{\boldsymbol{\phi}}(\boldsymbol{x}, \boldsymbol{z})$ is achieved, the marginals also matches which indicates the optimal $(\boldsymbol{\theta}^*, \boldsymbol{\phi}*)$ is achieved if and only if $p_{\boldsymbol{\theta}^*}(\boldsymbol{x}, \boldsymbol{z}) = q_{\boldsymbol{\phi}^*}(\boldsymbol{x}, \boldsymbol{z})$.

# B  Model Architecture

The model architectures are shown as following. For $f_{\boldsymbol{\psi}_1}(\boldsymbol{x}, \boldsymbol{z})$ and $f_{\boldsymbol{\psi}_2}(\boldsymbol{x}, \boldsymbol{z})$, we use the same architecture but the parameters are not shared.

Figure 1: Model architecture for MNIST

Figure 2: Model architecture for CIFAR

## $p_\theta(x|z)$

64 × 64 RGB images

5 × 5 conv. 32 stride 2 BN ReLU

3 × 3 conv. 32 BN ReLU

3 × 3 conv. 32 BN ReLU

3 × 3 conv. `64 stride 2 BN ReLU

3 × 3 conv. 64 BN ReLU

3 × 3 conv. 64 BN ReLU

3 × 3 conv. 128 stride 2 BN ReLU

3 × 3 conv. 128 BN ReLU

3 × 3 conv. 128 BN ReLU

3 × 3 conv. 256 stride 2 BN ReLU

3 × 3 conv. 256 BN ReLU

3 × 3 conv. 256 BN ReLU

3 × 3 conv. 256 BN ReLU

MLP — $\mu_z$

MLP — $\log \sigma_z^2$

## $q_\phi(z|x)$

2048 features

MLP

3 × 3 deconv. 256 stride 2 BN ReLU

3 × 3 deconv. 256 BN ReLU

3 × 3 deconv. 256 BN ReLU

3 × 3 deconv. 128 stride 2 BN ReLU

3 × 3 deconv. 128 BN ReLU

3 × 3 deconv. 128 BN ReLU

3 × 3 deconv. 64 stride 2 BN ReLU

3 × 3 deconv. 64 BN ReLU

3 × 3 deconv. 64 BN ReLU

3 × 3 deconv. 32 stride 2 BN ReLU

3 × 3 deconv. 64 BN ReLU | 3 × 3 deconv. 64 BN ReLU

3 × 3 deconv. 64 BN ReLU | 3 × 3 deconv. 64 BN ReLU

3 × 3 deconv. 64 BN ReLU | 3 × 3 deconv. 64 BN ReLU

$\mu_x$ | $\log \sigma_z^2$

Figure 3: Encoder and decoder for ImageNet

## $f_\psi(x, z)$

32 × 32 RGB images

5 × 5 conv. 32 stride 2 BN ReLU

5 × 5 conv. 64stride 2 BN ReLU

5 × 5 conv. 128 stride 2 BN ReLU

5 × 5 conv. 256 stride 2 BN ReLU

global average pooling

2048 features

MLP

concatenate

MLP

Figure 4: Discriminator for ImageNet

# C    Additional Results

Figure 5: Generated samples trained on CIFAR-10.

Figure 6: Generated samples trained on ImageNet.