[Reviews · NeurIPS 2017]

Reviewer 1



The paper proposes a variant of the Variational Auto-Encoder training objective. It uses adversarial training, to minimize a symmetric KL divergence between the joint distributions of latent and observed variables p(z,x)=p(z)p_\theta(x|z) and q(z,x)=q(x)q_\phi(z|x) . The approach is similar to the recent [ Mescheder, Nowozin, Geiger. Adversarial variational bayes: Unifying variational autoencoders and generative adversarial networks, 2016 ] in its joining VAE and GAN-like objective, but it is original in that it minimizes a symmetric KL divergence (with a GAN-like objective), which appears crucial to achieve good quality samples. It is also reminiscent of ALI [ Dumoulin et al. Adversarially learned inference, ICLR, 2017. ] as it attempts to match joint distributions. Experiments convincingly show that the approach manages to bridge to a large degree the gap in sampling quality with that of GANs, while at the same time achieving a very good tractable log-likelihood bound. Illustrations of ALI's comparatively poor reconstructions is also an interesting point. All this makes this research, in my eyes, a worthy and significant contribution. The main weakness of the paper to me is that I found the flow of the mathematical argument at times hard to follow. For ex.: - the role and usefulness of Lemma 1 and Corollary 1.1 appears much later than where they are introduced. - It is not shown that Eq 8 corresponds to a lower bound on the log-likelihood (as Eq 1 is known to be); is it or is it not? The argumentation of l 144 to 148 feels a little convoluted and indirect. I agree that Eq 8 is maximized when p(x,z) and q(x,z) match, but it is unclear to me, and from the paper, what the tradeoff between its two terms achieves when q does not have the capacity to match p. This should be discussed. l 183: Proposition 1. "equilibrium … is achieved" About GAN objectives you mentioned earlier in l 119 that "This objective mismatch may lead to the well-known instability issues associated with GAN training". Now your objective in Eq 12 uses a similar min-max objective. Couldn't there be similar instability or collapse issues? l 238: experimental evaluation of NLL, "estimated via the variational lower bound"; is this with the traditional VAE variational lower bound of Eq 1 ? Or with some later equation? In several places, you specify + const. It would be useful to briefly afterwards state what that const is and that it is constant w.r.t. what, as this is not always clear. There are a number of errors in some of the equations: l 145: "E_p(z) log p_\theta(z) = " should be "E_p(z) log q_\phi(z) = " Eq 10: I think $log q_\phi(x|z)$ should be $log q_\phi(z|x)$ Eq 11: has the term ordering (hence the sign) reversed in comparison to Eq 10. It should be changed to the same terms ordering as Eq 10 (to be consistent with Eq 10 and Eq. 12). l 150,152: "is not possible, as it requires an explicit form for", not only that, log p_\theta(x) is intractable.

Reviewer 2



This paper presents a method which aims to "symmetrize" the standard Maximum Likelihood (ML) objective. The approach is based on approximately minimizing both KL(q(x,z)||p(x,z)) and KL(p(x,z)||q(x,z)), where q(x,z) and p(x,z) are the "encoder" and "generator" distributions. The authors perform this feat using a mix of VAE and GAN techniques. The basic idea is to extract some of the log ratios in the KL divergences from "discriminators" trained on a couple of logistic regression problems. This leverages the fact that the optimal solution for logistic regression between distributions p and q is a function f such that f(x)=log(p(x)/q(x)) for all x. The motivation of mitigating the "over-dispersion" which comes from ML's focus on KL(q||p) is reasonable. ML does not penalize the model heavily for generating data where none exists in the ground truth, and many parts of the prior p(z) may not be trained by the regular VAE objective when KL(q(z|x)||p(z)) is large in training (which it typically is for images). The untrained parts of p(z) may produce bad data when sampling from the model. I'm not confident that the log likelihoods reported for MNIST, CIFAR10, and ImageNet were measured according to the protocols followed in previous work. The description of the MNIST experiments does not seem to match previous papers, which rely on explicitly binarized data. Also, the best published result on this dataset is 78.5 nats, from "An Architecture for Deep, Hierarchical Generative Models" (Bachman, NIPS 2016). The result reported for AS-VAE-r on CIFAR10 is surprisingly good, considering the architecture described in the supplementary material. The best "VAE only" result on this task is 3.11 from the most recent version of the IAF paper by Kingma et al. The IAF number required significant effort to achieve, and used a more sophisticated architecture. The 3.32 bpp reported for AS-VAE in the submitted paper is also a bit surprising, given the quality of reconstructions shown in Figure 3. The qualitative results in Figures 1-4 do not seem to improve on previous work. --Minor Corrections-- 1. $\log q_{\phi}(x|z)$ in Eqn 10 should be $\log q_{\phi}(z|x)$. 2. $\log p_{\phi}(z)$ on line 217 should be $\log q_{\phi}(z)$.

Reviewer 3



# Summary Motivated by the failure of maximum-likelihood training objectives to produce models that create good samples, this paper introduces a new formulation of a generative model. This model is based on a symmetric form of the VAE training objective. The goal of this symmetric formulation is to learn a distribution that both assigns high probability to points in the data distribution and low probability to points not in the data distribution (unlike a maximum-likelihood objective which focuses on learning distributions that assign high probability for points in the data distribution). As the resulting model is not tractable, the authors propose to estimate the arising KL-divergences using adversarial training. # Clarity This paper is mostly well-written and easy to understand. # Significance This paper describes an interesting approach to generative models, that attempts to fix some problems of methods based on maximum-likelihood learning. The proposed method is interesting in that it combines advantages from ALI/BiGans (good samples) with properties of VAEs (faithful reconstructions). However, by using auxiliary discriminators, the approach is only a rough approximation to the true objective, thus making it difficult to interpret the log-likelihoods estimates (see next section). The authors therefore should have evaluated the reliability of their log-likelihood estimates, e.g. by using annealed importance sampling. # Correctness === UPDATE: this concern was succesfully addressed by the authors' feedback === While the derivation of the method seems correct, the evaluation seems to be flawed. First of all, the authors evaluate their method with respect to the log-likelihood estimates produced by their method which can be very inaccurate. In fact, the discriminators have to compare the distributions q(x,z) to q(x)q(z) and p(x,z) to p(x)p(z) respectively which are usually very different. This is precisely the reason why adaptive contrast was introduced in the AVB-paper by Mescheder al. In this context, the authors' statement that "AVB does not optimize the original variational lower bound connected to its theory" (ll. 220-221) when using adaptive contrast is misleading, as adaptive contrast is a technique for making the estimate of the variational lower bound more accurate, not an entirely different training objective. It is not clear why the proposed method should produce good estimates of the KL-divergences without such a technique. For a more careful evaluation, the author's could have run annealed importance sampling (Neal, 1998) to evaluate the correctness of their log-likelihood estimates. At the very least, the authors should have made clear that the obtained log-likelihood values are only approximations and have to be taken with a grain of salt. It is also unclear why the author did not use the ELBO from equation (1) (which has an analytic expression as long as p(z), q(z | x) and p(x | z) are tractable) instead of L_VAEx which has to be approximated using an auxiliary discriminator network. If the authors indeed used L_VAEx only for training and used the ELBO from (1) (without a discriminator) for evaluation, this should be clarified in the paper. # Overall rating While the technique introduced in this paper is interesting, the evaluation seems to be flawed. At the very least, the authors should have stated that their method produces only approximate log-likelihood estimates which can therefore not be compared directly to other methods. # Response to author's feedback Thank you for your feedback. The authors addressed my major concern about the evaluation. I therefore raised the final rating to "6 - Marginally above acceptance threshold". Note, however, that AR #2 still has major concerns about the correctness of the experimental evaluation that should be addressed by the authors in the final manuscript. In particular, the authors should show the split of the ELBO for cifar-10 into reconstruction and KL-divergence to better understand why the method can achieve good results even though the reconstructions are apparently not that good. The authors should also make sure that they performed the MNIST-experiment according to the standard protocol (binarization etc.).